# A hybrid simulation model to study the impact of combined interventions on Ebola epidemic

**Peiyu Chen**[ID]*, **Wenhui Fan, Xudong Guo**

Department of Automation, Tsinghua University, Beijing, China

* cpy19@mails.tsinghua.edu.cn

## Abstract

Pandemics have been recognized as a serious global threat to humanity. To effectively prevent the spread and outbreak of the epidemic disease, theoretical models intended to depict the disease dynamics have served as the main tools to understand its underlying mechanisms and thus interrupt its transmission. Two commonly-used models are mean-field compartmental models and agent-based models (ABM). The former ones are analytically tractable for describing the dynamics of subpopulations by cannot explicitly consider the details of individual movements. The latter one is mainly used to the spread of epidemics at a microscopic level but have limited simulation scale for the randomness of the results. To overcome current limitations, a hierarchical hybrid modeling and simulation method, combining mean-field compartmental model and ABM, is proposed in this paper. Based on this method, we build a hybrid model, which takes both individual heterogeneity and the dynamics of sub-populations into account. The proposed model also investigates the impact of combined interventions (i. e. vaccination and pre-deployment training) for healthcare workers (HCWs) on the spread of disease. Taking the case of 2014-2015 Ebola Virus Disease (EVD) in Sierra Leone as an example, we examine its spreading mechanism and evaluate the effect of prevention by our parameterized and validated hybrid model. According to our simulation results, an optimal combination of pre-job training and vaccination deployment strategy has been identified. To conclude, our hybrid model helps informing the synergistic disease control strategies and the corresponding hierarchical hybrid modeling and simulation method can further be used to understand the individual dynamics during epidemic spreading in large scale population and help inform disease control strategies for different infectious disease.

## Introduction

The rapid spread of an infectious disease can have a devastating impact on humanity, lowering the quality of people's lives and causing increased mortality. To effectively reduce the epidemic size, considerable studies have been carried out to investigate the mechanism of the infectious disease dynamics and making optimal interventions for controlling epidemic size.

**Data Availability Statement:** All relevant data are within the paper and S1 File.

**Funding:** The author(s) received no specific funding for this work.

**Competing interests:** The authors have declared that no competing interests exist.

Mathematical models are tools to study the underlying mechanisms of infectious diseases and to evaluate the impact of interventions towards disease control [1–4]. Susceptible-Infected-Recovered (SIR) model, raised by Kermack and McKendrick [5], is the first epidemic model describing the dynamics of virus transmission among populations. The SIR model, a mean-field compartmental stochastic model, describes the individual mobility at a macroscopic level by separating the epidemic dynamics of sub-populations in patches and connecting them via transmission channels [6]. This model was further developed by Legrand J et al. by proposing a stochastic Susceptible-Exposed-Infected-Removed (SEIR) model to fit data from the 1995 and 2000 outbreaks in Congo and Uganda [7]. Furthermore, Berge et al. [8] studied the effect of vaccination and self-protection measures in the transmission dynamics of Ebola in Africa. However, the mean-field compartmental model cannot consider the stochastic nature of the movement of the individual explicitly for it cannot describe the mobility dynamics on individual level.

One the other hand, the ABM has been widely used to study the epidemic dynamics on the individual level which considers the individual activities such as birth, death, buried, infection, recovery and movement [9–11]. ABM can and has been used to examine how multi-level policies and programs influence population health [9]. Siettos et al. [12] developed an ABM with 6 million individuals interacting through a small-world social network in order to study the EVD in Sierra Leone. However, the randomness of ABM's output leads to the uncertainty in calibrating parameters and the sophisticated contact network. Therefore, when modeling large scale population, it is tough to gain an insight into the epidemic dynamics through ABM.

Countermeasures for disease prevention such as aiming at preventing disease transmission have been widely studied in recent years. Some researchers investigated single-focus intervention [13–15] while other works analyzed combinations of interventions [16–19]. Berge T. et al. [19] studied the impact of contact tracing, quarantine and hospitalization on disease dynamics and assessed the efficiency of different given control strategies. Hollingsworth et al. [17] investigated how contact-reducing interventions and availability of antiviral drugs or vaccines contribute to the epidemic dynamics. Zhang et al. [18] modeled and evaluated the spread of epidemic with intervention strategies of workforce shift and combination with school closure. With the increased complexity of interventions, the individual behaviour plays an essential role in the epidemic dynamics which cannot be treated by deterministic compartmental models assuming the sub-populations as well-mixed population. However, such a model is yet to be established and this challenge has motivated our study.

The unprecedented outbreak of Ebola virus disease(EVD), a significant public health problem, in West Africa during 2014–2016 led to 28,638 cases, with 11,316 deaths as of 20 January 2016 [20]. The World Health Organization(WHO) declared EVD a "public health emergency of international concern" in 2014 [21, 22]. The current EVD outbreak in West Africa is unprecedented in many ways, including the high number of doctors, nurses, and other health workers who have been infected. Siettos C. et al. [23] developed an ABM to investigate the epidemic dynamics of EVD and provided estimates for key epidemiological variables. Merler S. et al. [24] studied the influence of safe burials procedures, availability of Ebola treatment units and household protection kits by ABM. A new WHO report, investigating the infection of health worker(HCW) has indicated that HCWs are between 21 and 32 times more likely to be infected by Ebola than the general population [20]. In many epidemics, Healthcare worker (HCW) has been the transmission link to the general population, acting as 'super-spreaders' in early epidemics. As they belong to a high-risk and relatively easily identified group, it is necessary to evaluate the impact of healthcare worker-targeted intervention strategies. Vaccination is known as one of the most effective interventions for reducing the morbidity and mortality during the epidemic outbreak. Multiple mathematical modeling analyses aimed to evaluate the

impact of vaccination during the outbreak of Ebola epidemics have been undertaken [25–29]. Based on their study [24], Merler S. et al. [29] further developed a spatially explicit microsimulation model in order to study the effectiveness of ring vaccination strategies. Besides, predeployment training for HCWs also plays an important role in reducing infections. Studies indicate that the low level of knowledge, negative attitude and sub-standard practices can be eliminated through continued training and provision of needed and adequate resources in HCW's line of duties [30, 31]. However, few studies have incorporated both prophylactic vaccination strategies and the pre-deployment training strategies into the Ebola dynamic model.

In this paper, we propose a new hierarchical hybrid modeling and simulation method, coupling SEIR dynamic model and ABM, which takes the advantages of each approach (individual heterogeneity for ABM and population size and reliable output for SEIR model) into consideration. Based on the characteristic of Ebola virus transmission in West Africa, we consider the problem of how to study the epidemic dynamics in the entire region while the individual heterogeneity is also considered. We develop a hybrid model based on the hierarchical structure to simulate the dynamics of Ebola epidemic and evaluate the impact of intervention strategies. Different vaccination and training strategies targeted to HCWs are incorporated into the proposed hybrid model so as to provide optimal interventions. The published data by WHO for 2014–2015 epidemic in Sierra Leone are used for validation and the recommended vaccination and training interventions are given for the prevention of the spread of disease.

Our work makes three contributions. Firstly, we propose a hierarchical hybrid modeling and simulation method which constructs infectious disease transmission dynamics at different scales. The "macroscopic" model (based on SEIR model) describes the size of the general population in the affected area and the "microscopic" model (based on ABM) describes how multi-level policies and programs shape population health. Secondly, we build an extended SEIR dynamic model based on the newly raised hierarchical modeling method and take into account both vaccination and training strategies on HCWs to better understand the dynamics of Ebola epidemic and its treatment. Thirdly, we reveal the essential characteristic of the impact of combined interventions on EVD transmission, extending the previous work by taking synergistic strategies into account.

The content of this paper is organized as follows. "Methods" describes the proposed hierarchical hybrid modeling and simulation method and its corresponding hybrid model analyzing the dynamics of disease transmission during the 2014 EVD outbreak in Sierra Leone. "Results" presents the validation results of our proposed model and studies the effects of combined interventions. "Discussion" illustrates our conclusion, the comparison between the previous work corresponding to intervention strategies and our hybrid model and the future work based on our present study.

## Methods

### Model data

The epidemic of Ebola in Sierra Leone was simulated and validated using both specific disease characteristic parameters derived from literature researches and the parameters estimated by simultaneously minimizing the squared differences between the model output of infectious cases and the cases reported by WHO for the 2014 epidemic as reported [32, 33]. The simulation period (from May 12, 2014 to November 13, 2015) is divided into three phases (first period: May 12, 2014 to September 12,2014 a total of 123 days, second period: September 12,2014 to November 18,2014, a total of 67 days, third period: November 19,2014 to November 13, 2015, a total of 360 days) according to the growth trend of reported cases. The reported data is shown in Fig 1. In the first 123 days, a sharpen increase in the reported Ebola cases

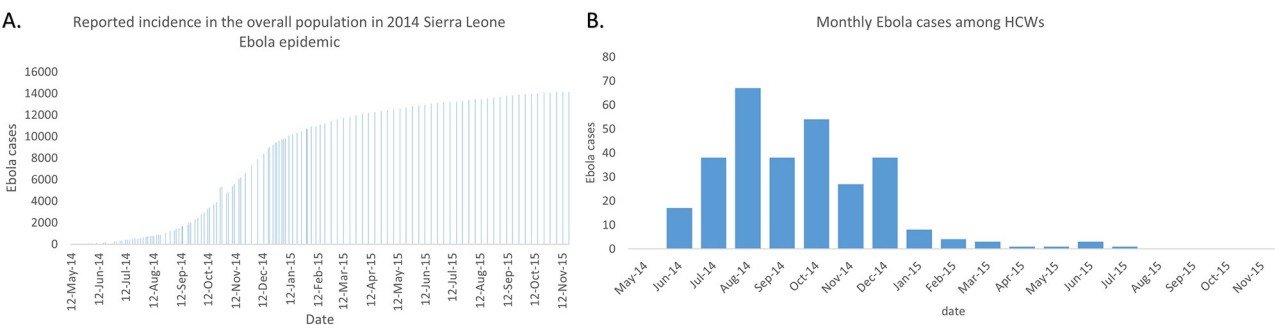

**Fig 1. Reported Ebola cases for 2014–2015 Sierra Leone epidemic among overall population and among HCWs.** Data for the overall population cases were reported by WHO [32] and data for HCW cases were collected by Fang et al. [33]. A: Reported incidence in the overall population in 2014–2015 Sierra Leone epidemic(cumulative cases). B: Weekly Ebola cases among HCWs(weekly cases).

among HCWs per month and a slight increase per week in reported cases among overall population are shown. In the second period (123 to 190 days) the new cases among HCWs decreased and the new cases among the overall population show rapid increase. In the third period (190 to 550 days), both cases among HCWs and the overall population decrease (Fig 1).

## Model assumption

The main purpose of this model is to simulate the outbreak of Ebola and analyse the influence of different education and vaccination scenarios. Considering the real affection of vaccination and education, we makes the following assumptions:

1. Population births, deaths from other factors(except caused by Ebola) during the epidemic are not considered.

2. Vaccine assumed: time to onset of protection: 7 days. Duration of protection: 180 days. Efficacy: 100%. [34] Daily rate at which vaccination is carried out: 5% HCW.

3. Only trained HCW have opportunities to be vaccinated.

4. There are several phases in the education and the probability of being infected correlates with HCW's level of education.

5. Education assumed: pre-deployment training is prior to HCWs who are willing to be trained.

## Hybrid modeling and simulation method and the model

Traditional epidemiological models are 'susceptible, exposed, infected, and removed (SEIR)' model or its variants. SEIR model or its variants picture the epidemic dynamics inadequately as continuous deterministic processes and strongly simplified the representation of social activities. However, considering the impact of intervention strategies in the emergency response, the disease transmission dynamic model needs to add up details for individual activities. We propose a hybrid modeling and simulation method combining ABM and SEIR dynamic. In our method, epidemiological model based on SEIR framework is used at the macro level, inside which agents are modeled at the micro level using ABM. The mobility dynamic of sub-populations is described on the macro level while the individual heterogeneity is shown in the micro level. An illustration of the ABM method based on SEIR in the hybrid model is presented in Fig 2. Our method handles stochastic events such as vaccination and

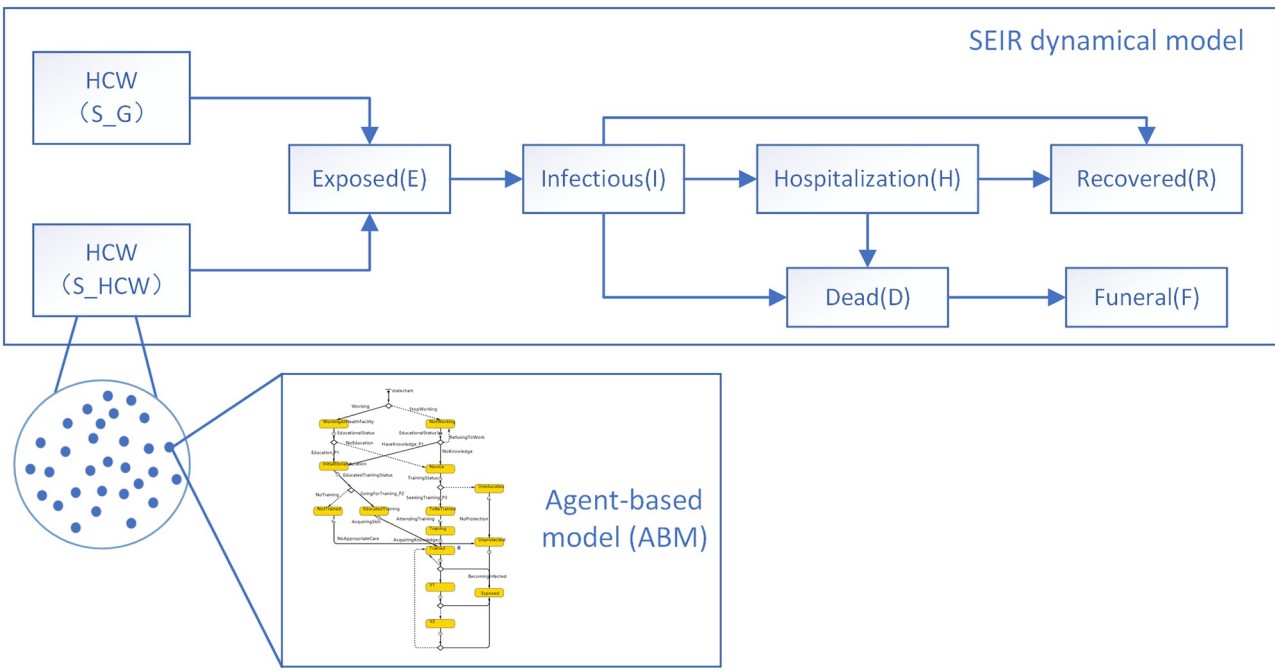

**Fig 2. The whole frame of the hybrid model.** $S_{HCW}$ means susceptible HCW and $S_G$ means susceptible general population. The "susceptible HCW" sub-population acts as linkage between the agent-based modeling and the system dynamics modeling. The state transition diagram shown in the ABM has a detailed description in section "Agent-based modeling".

pre-deployment training and continuous processes like the spread of virus at the same time. The global model runs in hybrid time with epidemic dynamics calculated in continuous time, and individual state changes and their related processes (training, vaccination stage, decision to work) occurring as discrete asynchronous events.

Based on this hybrid modeling and simulation method, we build a detailed hybrid model to investigate the epidemic dynamics of EVD in Sierra Leone, study the effect of interventions (vaccination and education) on HCW and provide recommendations for potential future health-related emergency responses for the government. The hybrid model can reproduce the viability of a HCW in a heterogeneous environment by coupling a decision making submodel in ABM, with a contact infection model in SEIR model.

**SEIR dynamic model.** According to [25], the study expend the general SEIR model which separates susceptible group into susceptible HCW and susceptible general population. In our study, we introduce a related but also new model with a difference. In our SEIR model, we use a similar structure as [25]. Eight compartments in the model, as shown in Fig 2, categorize the total population in Sierra Leone. The population in each individual compartment at any given point in time has been estimated by accounting for the inflows and outflows into the compartment. In addition to the general SEIR model, the sub-population "susceptible(S)" are separated into "susceptible HCW" and "susceptible general population" as shown in Fig 2. In addition, the "susceptible HCW" has been modeled in a deeper level by ABM and its detail is described in "Agent-based model" part.

Briefly, Susceptible individuals (S) may be Exposed (E) to virus after contact with an infectious individual and become Infectious (I) after the incubation of virus, being capable of infecting others. Some of the infected individuals may be Hospitalized (H). The unhospitalized patients in I and the hospitalized patients in H will be in one of two states: be dead(D), with

capacity of infecting others before going through safe burials procedures or holding funerals (F), or they may recover(R). The transmissions and the ordinary differential equations describing this model are listed in the S1 File.

**Agent-based model.**   ABM is a microscopic modeling method and its benefits can be summarized as followed: i. ABM can describe emergent phenomena; ii. ABM describes system in a natural way; and iii. ABM's flexibility. In our study, multiple interventions such as education and vaccination need to be considered [35]. The HCW's state of change is nonlinear and can be depicted by chance or if-then rules. It is difficult to describing discontinuity in individual behavior with aggregate flow equations. Therefore, ABM is used for its ability to deal with such system. Besides, ABM describes and simulates our system in a natural way. In our ABM, each parameter had intuitive meaning and the adjustment of the value has guiding significance to government policy making. We do not generate social-transmission network in ABM for we only focus on the uninfected HCWs who do not contribute to virus transmission and the contact infection is considered in the SEIR model.

In order to study the effect of interventions on HCWs, we assume the susceptible HCWs ($S_{HCW}$) as "agents" in order to study their behavior towards the given interventions. As in Fig 2, ABM is applied to gain an insight into the group of $S_{HCW}$ in the SEIR dynamic model. Each agent has an associated social state related to their reaction towards education or vaccination interventions. The agent makes decisions depending on its current condition. The output of the ABM is the number of exposed HCWs calculated by the information gathered in the SEIR model. And the exposed HCWs, in return, join in the dynamics of virus transmission in the SEIR model.

The micro level model structure is shown in Figs 3–5. Considering the vaccination and education strategies, our micro level model is separated into two modules: education and vaccination. The decision tree is depicted by a state-chart diagram built inside the model and evaluated once per time step. According to the transition rules, each HCW changes his or her state by time step. (S1 File).

In the education module, HCWs are separated into several types according to their academic certificate and their willingness to work. The arrows in Fig 3 represent the possible transitions and their shifting mechanism follows the rules presented in the earlier study [36]. We mainly focus on the pre-deployment training which prepared for HCW in the Ebola affected area. Other factors which represent the situation of education are also considered in the ABM. We specifically designate the parameters that represent the amount of schooling (professional training for HCWs) as several decision making processes (diamond shapes in Fig 3) and their corresponding transition rate P1-P3. The meaning and value of these three parameters are described in Table 2. Here, we focus on the rate of P3 which explains the coverage of pre-deployment training among HCWs.

At the first stage of the state chart(Fig 3), those who had knowledge of EVD before the outbreak turn into "Initial Ebola Education" state while others turn into the state of "Novice". At the second stage of the state chart, those who have knowledge of EVD before the outbreak attend intensive training according to the transmission rate "P2" while those who have low level of knowledge attend the pre-deployment training according to the transmission rate "P3". After these two stages, those who do not attend the training would be "unprotected" and eventually come to the "Exposed" state after a incubation period of 2 to 21 days of the Ebola virus while working in health facilities. Those who attend the training move to the "trained" state and thus carry out appropriate precaution and take the initiative to get vaccination.

HCWs who have been intensively trained (HCWs who were in the "Trained" state) move into the vaccination module (Fig 4) according to the vaccine penetration rate. The two

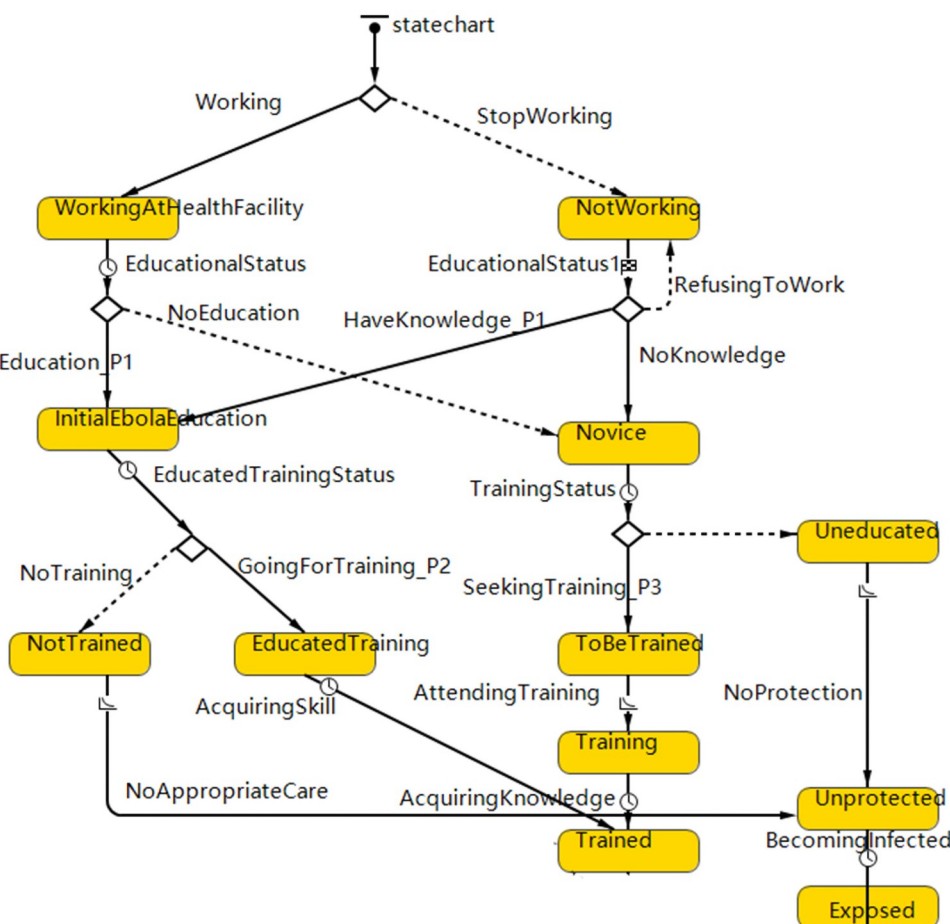

**Fig 3. The education module in the agent-based model.** The state chart presents each HCW's behavior towards education interventions. The rectangular boxes represent the states of HCWs, the diamond means branch, the solid arrow represents transition under a certain condition (ex. "Yes"), and the dashed arrow represents the default (ex. "No").

modules are connected though the state of "trained" as shown in Fig 5. As mentioned in model assumption, only the "trained" HCWs can further turn into the vaccination state. Considering the vaccine characteristic which includes the onset of protection and the duration of protection, agents in this phase are separated into several states: i. HCWs who have already vaccinated but the efficacy of vaccine has not been onset yet (V1), ii. HCWS who are protected by the effective vaccine (V2), iii. Vaccinated HCWs revert to susceptible state after the efficiency of vaccine (remain in the state "Trained"). The transformation rules of these three states follow the vaccine characteristics. The main transition paths are listed as followed:

1. HCWs in "trained" state move to "V1" according to the vaccine penetration.

2. HCWs in "trained" state turn to be "Exposed" before getting vaccination.

3. HCWs in "V1" state turn to be "Exposed" before the onset of the efficacy of vaccine.

4. HCWs in "V1" state move to "V2" as the vaccine starts taking effect.

5. HCWs in "V2" state revert to "Trained" state upon expiration of vaccine efficacy.

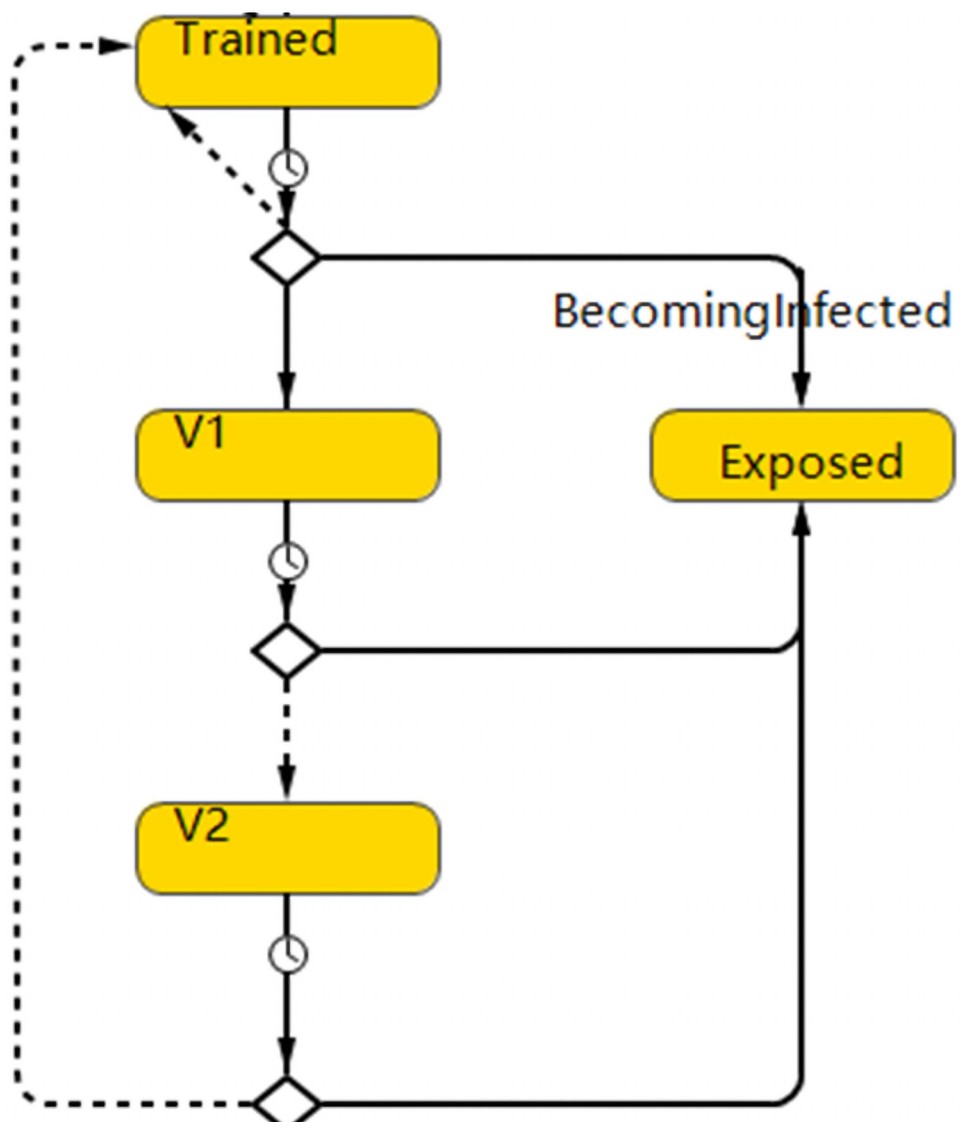

**Fig 4. The vaccination module in the agent-based model.** "V1" represents HCWs who have already vaccinated but the efficacy of vaccine has not been onset yet. "V2" means HCWs who are protected by the effective vaccine.

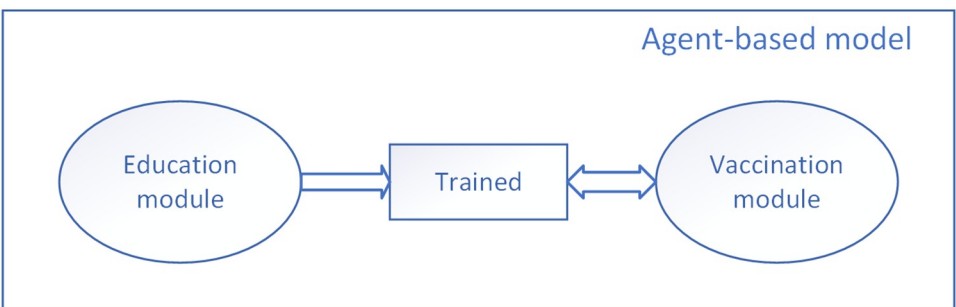

**Fig 5. The structure of the model of micro level based on the activity of HCWs.** The education module and the vaccination module are concentrated by the state "Trained" which is described both in the two modules. The arrows demonstrate the available state transition path.

### Experimental design

In order to analyze the effects of implementation of both independent and synergistic interventions, scenes under independent intervention and synergistic interventions are considered.

1. Without interventions: The simulation runs with the parameters that are parameterized and validated by the data reported by WHO.

2. Pre-deployment training strategies: The impact of education strategies are assessed by raising the rate of P3 from 95% to 100% at a step of 1%. Additionally, the value of P3 keeps 95% in the scenario that without interventions.

3. Vaccination strategies: The strategies are evaluated by prophylactic vaccination of 10%, 30% and 50% of HCWs.

4. Combined intervention strategies: P3 varies from 95% to 100% at a step of 1% with 10%, 30% or 50% HCWs get vaccination.

The cumulative number of cases and deaths represents the size of the epidemic. The variation trend in the cumulative number of cases and deaths indicates the effect of interventions.

## Results

### Model parameterization and validation

The parameters fitted to the published data by WHO for 2014–2015 epidemic in Sierra Leone can be seen in Tables 1 and 2. Parameters in the ABM are fixed and their values are determined as close to reality as possible according to the prior knowledge [37–40]. Therefore, the free parameters ("Fitted" parameters listed in Table 1) that require calibration are in the SEIR dynamic model. We develop the hybrid model using the multi-method simulation software AnyLogic Professional (version 8.3.1) [41] and perform calibration by "Calibration Experiment" in AnyLogic. The daily new cases and deaths in the overall population were used for

**Table 1. Parameters for the hybrid model.**

| Parameter | Description | 0–123 Days | 123–190 Days | 190–550 Days | Source |
|---|---|---|---|---|---|
| N | Size of the total population in Sierra Leone (2014) | | 7,017,144 | | [42] |
| HCW | Health care worker | | 1153 | | [43] |
| $1/\sigma$ | Incubation Period | 7 days | 7 days | 7 days | [7] |
| $1/\gamma_D$ | Mean duration from death to burial | 2 days | 2 days | 2 days | [7] |
| $1/\alpha$ | Mean duration from onset of infection to hospitalization | 2.4 days | 2.4 days | 2.2 days | [25] |
| $\beta_{I_{HCW}}$ | Contact rate for infectious individuals in HCW | 117.8 | 15 | 5.1 | fitted |
| $\beta_{H_{HCW}}$ | Contact rate for hospitalized individuals in HCW | 189.21 | 23.64 | 8.88 | fitted |
| $\beta_{D_{HCW}}$ | Contact rate for dead but not buried individuals in HCW | 0.0726 | 0.0525 | 0.045 | fitted |
| $\beta_{I_{NHCW}}$ | Contact rate for infectious individuals in general population | 0.635 | 0.594 | 0.425 | fitted |
| $\beta_{H_{NHCW}}$ | Contact rate for hospitalized individuals in general population | 0.002 | 0.001 | 0.0009 | fitted |
| $\beta_{D_{NHCW}}$ | Contact rate for dead but not buried individuals in general population | 0.07 | 0.0514 | 0.048 | fitted |
| $\delta1$ | Case fatality rate for unhospitalized individuals | 0.46 | 0.38 | 0.19 | fitted |
| $\delta2$ | Case fatality rate for hospitalized individuals | 0.46 | 0.38 | 0.19 | fitted |
| $1/\gamma$ | Mean duration from onset of infection to death or recovery | 6 days | 6 days | 6 days | [25] |
| $1/\gamma_H$ | Mean duration from hospitalization to death or recovery | 6.2 days | 8.3 days | 16 days | [25] |
| K1 | Proportion of HCW among the total population at the start of the epidemic | | 0.016% | | Calculated |

Parameters that are not tagged with sources were fitted to the published data according to the basic model output.

**Table 2. Parameters for the hybrid model (education related).**

| Parameter | Description | Value | Source |
|-----------|-------------|-------|--------|
| P | The number of HCWs trained in a week | 100 HCWs | [38] |
| T | Duration of intensive training | 5 days | [37]. |
| NK | proportion of HCW who stopped working did not have knowledge of EVD | 60% | [37] |
| P1 | the percentage of the total HCWs that had knowledge of EVD before the outbreak | 10% | [38–40] |
| P2 | the percentage of HCWs who had knowledge of EVD before the outbreak and attended intensive training during the outbreak | 90% | [38–40] |
| P3 | the probability for unprepared and unskilled frontline HCW to attend pre-deployment training during the outbreak | 95% | [38–40] |

calibration. Through calibration experiment, the difference between the simulation output and the given data is calculated with the help of the "difference" function. The integration range is the intersection of argument ranges of the two datasets. The "difference" function returns a non-negative value which is a square root of the average of square of difference between sets of data. By running 3000 iterations of the calibration experiment, we obtain the minimal value and the corresponding parameters, as it means the least difference between two sets of data. In addition, we set fixed intervals for the free parameters during calibration referring to the parameters reported by Potluri R. [25].

The comparison between the model output of Ebola cases and deaths among total population and among HCWs in Sierra Leone with the data reported by WHO are shown in Fig 6. The deterministic model fit well for both the cases in the overall population and the cases among HCWs, for the curve of the model output is close to the curve of the reported cases in both groups. The starting date of the x-coordinate "day 0" represents March 22,2014 while the end of the simulation date represents November 13,2015.

The mean model outputs show 14452 cases(95% CI 8650–16310) and 4041(95% CI 2734–5851) deaths compared to 14122 cases and 3955 deaths reported by WHO. Considering the group of HCWs, the mean model outputs show 312 cases compared to 299 cases reported by WHO. The basic reproduction number R0 for this hybrid model is $1.30(1.26 \leq R0 \leq 2.53)$ which is consistent to the reported estimation of R0 for the 2014 Sierra Leone epidemic in the earlier work [27].

## Impact of pre-deployment training

To evaluate the impact of pre-deployment training, Fig 7 shows cumulative infected population under different coverage rate of pre-deployment training among HCWs. Additionally, although the value of P3 in the initial case may not precisely reveal the real situation, the bias of P3 has little impact on our study for we mainly focus on the trend rather than the specific value. It can be seen that i. a small increase of the number of HCW (95% to 96% HCW) getting pre-deployment training can help reduce 31.93% cases and 31.85% deaths among the overall population (Table 3). ii. About 10% of Ebola cases and deaths among general population can be reduced by 1% increase in the coverage of pre-deployment training for HCW. iii. Enhancing the education for the frontline HCW can have a positive impact on the disease prevention.

## Impact of vaccination

To evaluate the impact of vaccination, we compare the simulation results between the cases under different prophylactic vaccination strategies. Fig 8 shows the results of different

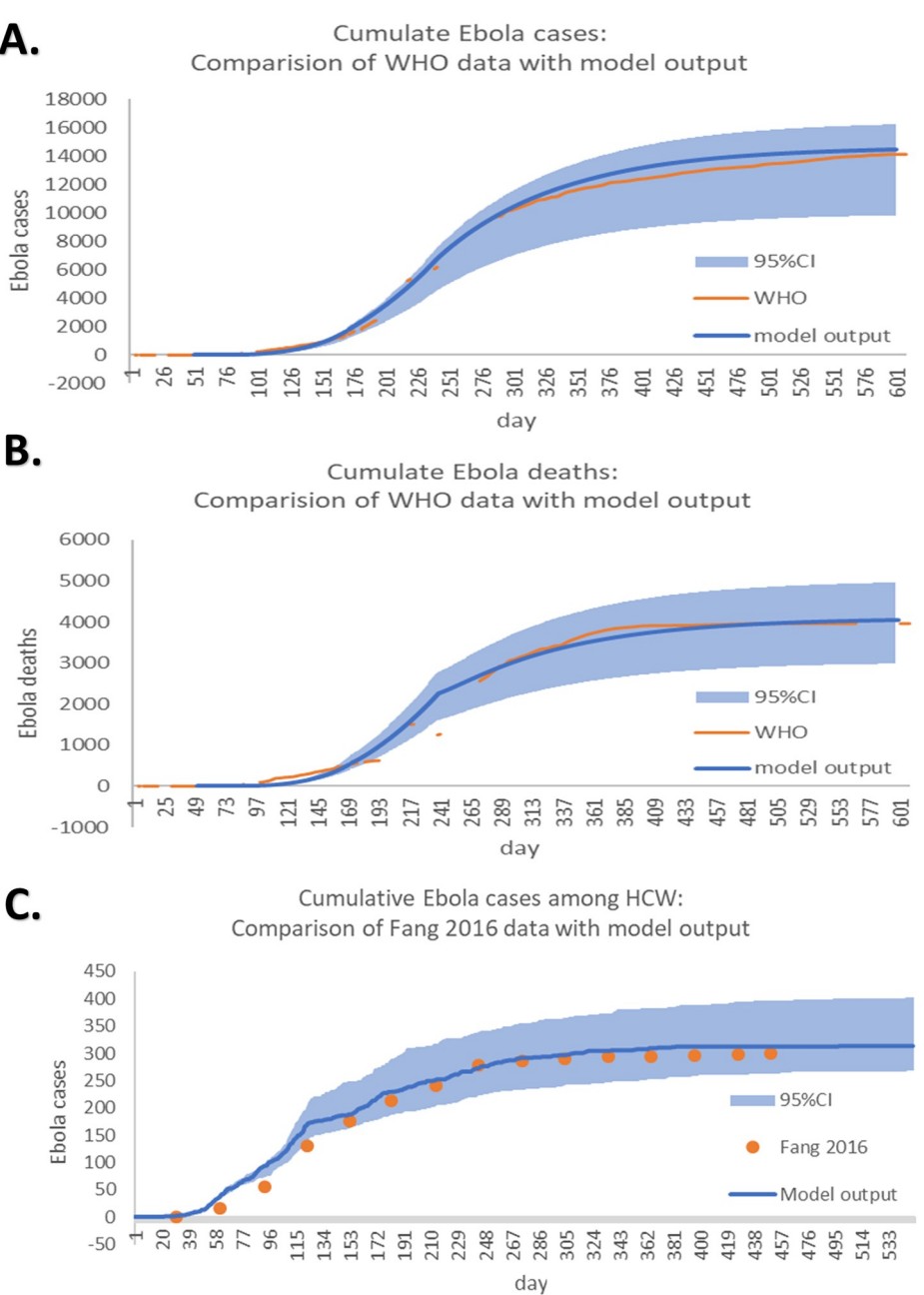

**Fig 6. Comparison of the output of the simulation result with published data.** A: Comparison of cumulative Ebola cases reported by the WHO with model output. B: Comparison of cumulative Ebola mortality reported by the WHO with model output. C: Comparison of cumulative HCW Ebola cases reported by Fang et al. with model output.

coverage rate of HCWs prophylactic vaccination and the cases without intervention. The model output shows that when 10% HCWs get vaccination (about 115 HCWs which make up 0.0016% of the population in Sierra Leone), 54% cases and 51% deaths of the general population can be avoided. In other word, 67.33 cases can be averted by per vaccination when 10% HCWs get vaccination. When 30% HCWs get vaccination, 26.17 cases can be averted by per vaccination. Besides, vaccination of 50% HCW helps reduce 16.38 cases per vaccination (Table 4).

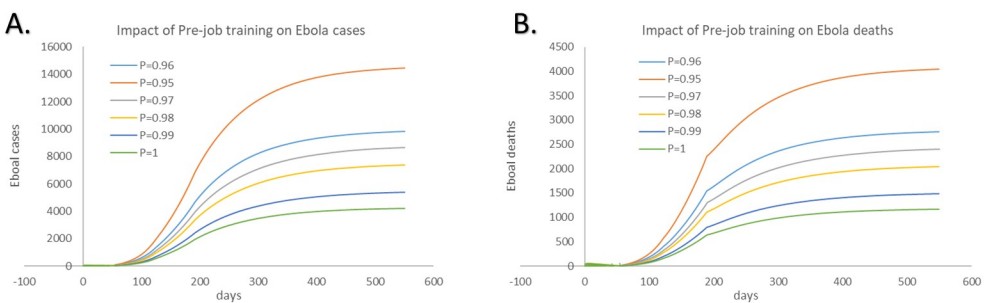

**Fig 7. Impact of pre-deployment training of different proportions of HCWs on cumulative Ebola cases and deaths.** A: Cumulative cases. B: Cumulative deaths.

**Table 3. Impact of pre-deployment training for healthcare workers on cumulative cases and deaths among overall population caused by EVD.**

| Parameter | 95% of HCW | 96% of HCW | 97% of HCW | 98% of HCW | 99% of HCW | 100% of HCW |
|---|---|---|---|---|---|---|
| Number trained | 1095 | 1107 | 1118 | 1130 | 1141 | 1153 |
| Cumulative cases | 14452 | 9838 | 8668 | 7360 | 5377 | 4206 |
| Proportion of cases averted vs no intervention | - | 31.93% | 40.02% | 49.07% | 62.79% | 70.90% |
| Cumulative deaths | 4041 | 2754 | 2398 | 2034 | 1480 | 1164 |
| Proportion of deaths averted vs no intervention | - | 31.85% | 40.66% | 49.67% | 63.38% | 71.20% |

The 95%CI for the output in the table are shown in S1 File.

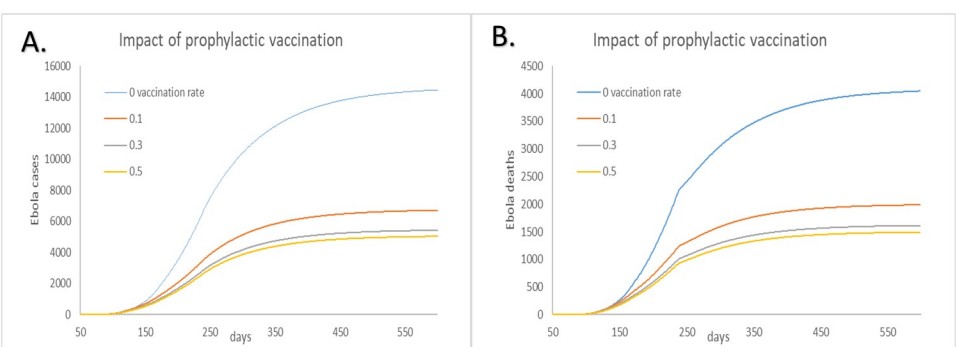

**Fig 8. Impact of prophylactic vaccination of different proportions of HCW on cumulative Ebola cases and deaths.** A: Cumulative cases. B: Cumulative deaths.

**Table 4. Impact of vaccination for healthcare workers on cumulative cases and deaths among overall population caused by EVD.**

| Parameter | No vaccination | 10% of HCW vaccinated | 30% of HCW vaccinated | 50% of HCW vaccinated |
|---|---|---|---|---|
| Number vaccinated | 0 | 115 | 345 | 575 |
| vaccination efficiency/per vaccination | - | 67.33 | 26.17 | 16.38 |
| Cumulative cases | 14452 | 6708 | 5423 | 5031 |
| Proportion of cases averted vs no vaccination | - | 53.58% | 62.48% | 65.19% |
| Cumulative deaths | 4041 | 1980 | 1607 | 1489 |
| Proportion of deaths averted vs no vaccination | - | 51.00% | 60.23% | 63.15% |

The 95%CI for the output in the table are shown in S1 File.

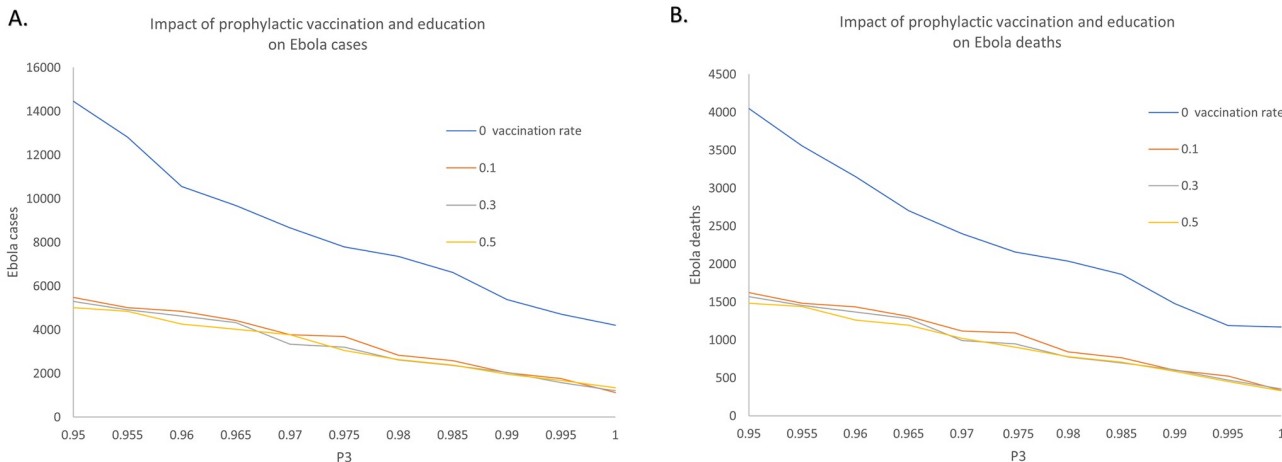

**Fig 9. Impact of combined interventions on cumulative Ebola cases and deaths.** The combined intervention refers to combination of education and vaccination strategies. A: Cumulative cases. B: Cumulative deaths.

The efficiency of vaccination also depends on the time estimated for completing vaccinations among certain group. However, the duration time of vaccinating the certain group is also influenced by the infrastructural challenges, the time the vaccination taken for onset of protection and also the degree of dispersion of the people who need vaccination.

Comparing to the previous study result that effective EVD vaccination rates were estimated to be 42% for the overall population [44], our study shows that only 10% HCWs (0.00016% of the whole population) get vaccination can be effective. The simulation results indicate that i. vaccination of a small proportion of populations can effectively reduce the size of epidemic. Also, vaccinating HCWs can be seen as an effective method to eliminate virus transmission channels between HCWs and the infected people. ii. The aversion rates in Table 4 show that vaccination of 30% and 50% HCW have similar effect. This result indicates that the effects of vaccination on HCW are limited with the expansion of vaccination coverage. iii. Considering the efficiency of vaccination, vaccination of 30% HCW is recommended.

### Impact of combination of vaccination and education

To explore the effects of the combined interventions, we vary the pre-deployment training rate from 95% to 100% at a step of 1% while the values of the vaccination coverage vary from 10% to 50% with increments of 20%. The simulation results which collected the epidemic cases and deaths of scenarios with respect to different intervention combinations is demonstrated in Fig 9. Fig 10 depicts the median, interquartile and 1.5 times the interquartile range of the output under different intervention strategies. To further investigate the effects of the segregation strategies, Fig 11 illustrates the simulation results by the aversion rate of cumulative incidence and mortality compared with the no intervention scenario.

Respectively, Tables 5 and 6 shows the cases, deaths as well as the proportion of cases averted versus the cases under no intervention scenarios, corresponding to the result in Figs 9 and 10. It can be seen that the number of cumulative cases decreases gradually with the increase of education rate under the same coverage proportion of vaccination, shown in Figs 9 and 10. However, the influence of vaccination under the synergistic interventions showed difference with the single intervention cases. The cumulative cases and deaths do not keep a remarkable decrease with the increase of vaccination rate. According to Table 5, with P3 kept the rate of 0.96, the proportion of cases averted are seen to be 31.93% without vaccination as

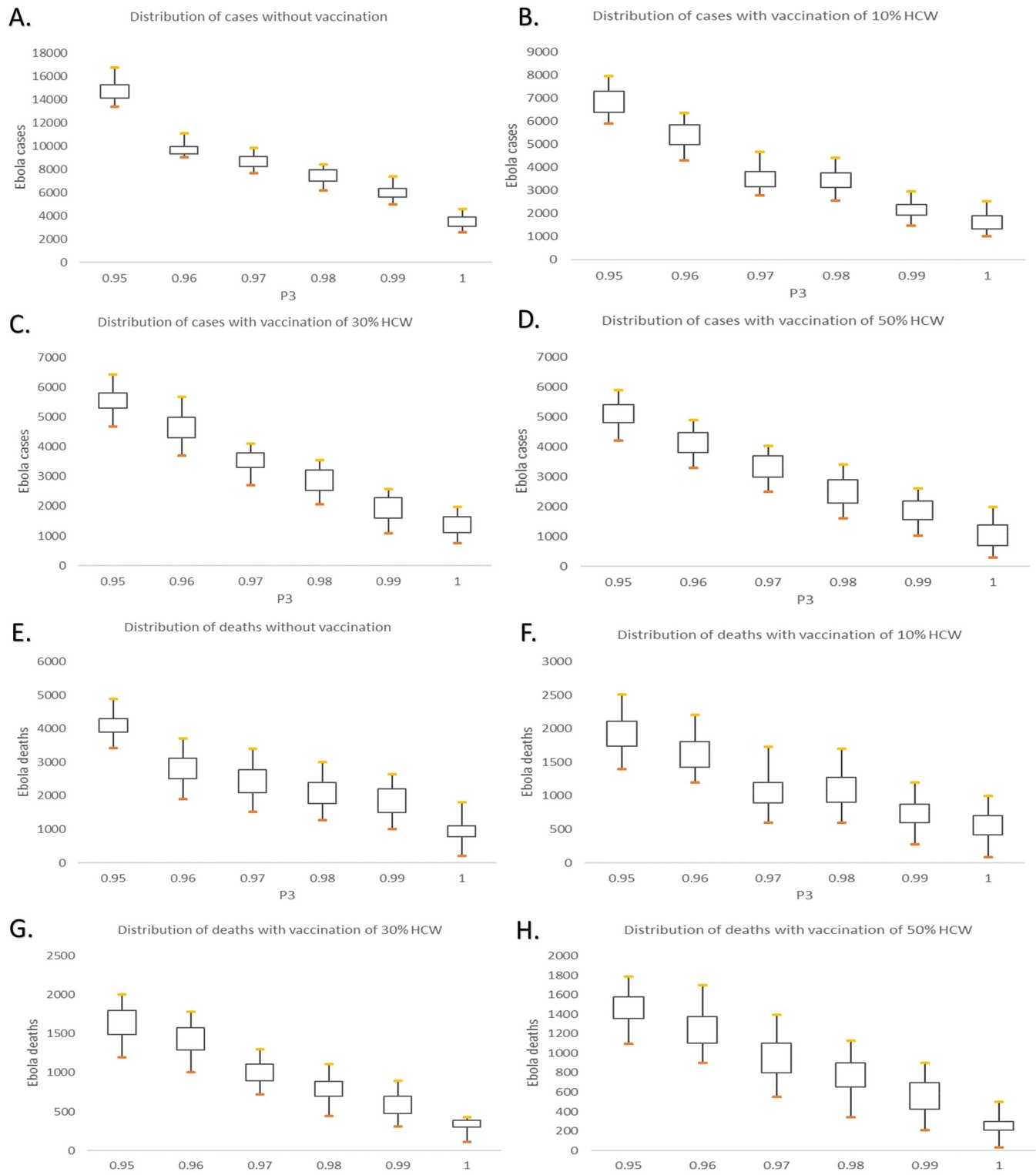

**Fig 10. The median, interquartile range and 1.5 times the interquartile range of the simulation output under different combination of interventions.** A-D: Cumulative cases under different combination of intervention strategies. E-H: Cumulative deaths under different combination of interventions.

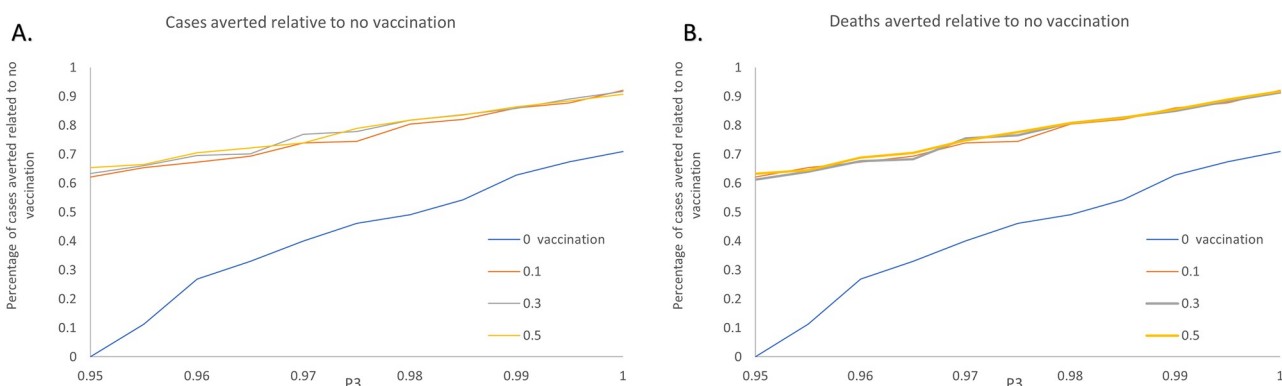

**Fig 11. Impact of combined interventions on aversion rate compared with no vaccination scenario.** A: Cases aversion rate. E-H: Deaths aversion rate.

compared to 67.27% with 10% of HCW getting vaccination. 35.34% cases averted by the intervention of increasing 10% HCW to get vaccination.

In contrast, vaccination of 50% of HCW with 96% HCW getting trained results in an aversion of 70.54% cases which means that only 3.27% cases can be averted by increasing 40% HCW to get vaccination. Comparing with the without education intervention scenarios, 53.58% of cases can be averted when 10% of HCW get vaccination while 62.48% of cases can be averted with 30% vaccination proportion. A 8.9% increase is seen between the '10% vaccination' scenario and '30% vaccination' scenario. With the further increase of the coverage of pre-deployment training, the downward shift in the overall Ebola cases and deaths becomes less pronounced among the vaccination rate of 10%, 30% and 50%.

Based on Fig 11, Tables 5 and 6, there are several conclusions to be made concerning these results. i. The combination of interventions for Ebola prevention and protection is more effective than the single intervention. ii. Although the combination of interventions is seen to add substantial benefit to the epidemic prevention and protection, the efficiency of expanding vaccination coverage shows a decline when the education strategies are implemented simultaneously. iii. Unlike the recommended strategies given in the "Impact of vaccination strategies", vaccination of 10% of HCW can achieve a higher efficiency than vaccination of 30% of HCW. In other word, with the intervention of expanding the coverage of pre-deployment training for

**Table 5. Impact of combined interventions on cumulative Ebola cases and the proportion of cases averted versus no vaccination scenarios.**

| Vaccination rate | P3 | | | | | |
|---|---|---|---|---|---|---|
| | **0.95** | **0.96** | **0.97** | **0.98** | **0.99** | **1** |
| 0 | 14452 | 9838 | 8668 | 7360 | 5377 | 4206 |
| Proportion of cases averted vs no vaccination | 0 | 31.93% | 40.02% | 49.07% | 62.79% | 70.90% |
| 0.1 | 5475 | 4730 | 3771 | 2837 | 2035 | 1131 |
| Proportion of cases averted vs no vaccination | 62.11% | 67.27% | 73.91% | 80.37% | 85.92% | 92.17% |
| 0.3 | 5291 | 4396 | 3345 | 2624 | 2045 | 1199 |
| Proportion of cases averted vs no vaccination | 63.39% | 69.58% | 76.85% | 81.84% | 85.85% | 91.70% |
| 0.5 | 5009 | 4258 | 3770 | 2625 | 1966 | 1345 |
| Proportion of cases averted vs no vaccination | 65.34% | 70.54% | 73.91% | 81.84% | 86.40% | 90.69% |

The 95%CI for the output in the table are shown in S1 File.

**Table 6. Impact of combined interventions on cumulative Ebola deaths and the proportion of deaths averted versus no vaccination scenarios.**

| Vaccination rate | P3 | | | | | |
|---|---|---|---|---|---|---|
| | **0.95** | **0.96** | **0.97** | **0.98** | **0.99** | **1** |
| 0 | 4041 | 2754 | 2398 | 2034 | 1480 | 1164 |
| Proportion of deaths averted vs no vaccination | 0 | 31.85% | 40.66% | 49.67% | 63.38% | 71.20% |
| 0.1 | 1622 | 1399 | 1115 | 841 | 601 | 333 |
| Proportion of deaths averted vs no vaccination | 59.86% | 65.38% | 72.41% | 79.19% | 85.10% | 91.76% |
| 0.3 | 1569 | 1308 | 992 | 774 | 603 | 353 |
| Proportion of deaths averted vs no vaccination | 61.17% | 67.63% | 75.45% | 80.85% | 85.08% | 91.26% |
| 0.5 | 1484 | 1261 | 1019 | 778 | 581 | 330 |
| Proportion of deaths averted vs no vaccination | 63.28% | 68.79% | 74.78% | 80.75% | 85.62% | 91.83% |

The 95%CI for the output in the table are shown in S1 File.

HCW, 10% of vaccination rate is recommended in the process of Ebola epidemic prevention and protection.

## Discussion and conclusion

Given that analysis mentioned above, we draw the following conclusions: i. The hybrid modeling and simulation method we propose in this work can be used to construct the hybrid simulation method model based on the specific epidemic case. ii. The hybrid simulation model fits the dynamics of 2014–2015 Ebola epidemic outbreak in Sierra Leone. iii. Our proposed hybrid model can simulate the implementation of synergistic interventions which effectively slow down the spread of epidemic. iv. Vaccination of 10% HCW in addition to intensified training on HCWs during the outbreak has a significant impact and the further increase of the vaccination rate results in little improvement.

Our model combines differential equations for the virus transmission among subpopulations in the epidemic affected area and ABM for the management of HCWs. Such muti-scale modeling method was also used in meta-population dynamics modeling [45]. In order to minimize the randomness raised by ABM, we build the ABM for a small group of people who are targeted to the interventions and have a significant impact on disease dynamics. Our model depicts the individual details selectively and keeps the structure of the SEIR model which ensures the ability to model in large scale population. Additionally, the number of HCWs be exposed plays the role of linking between the two modeling level. This model, engaging various dynamics at different scales, outlines a promising method for large-scale system modeling. The combination of SEIR model and ABM has many advantages: i. Intuitive dynamic representation by using data acquired in the real world and modeling at different scales (e.g., spatial scales and temporal scales). [46] ii. Modeling individual activities, rare events and the virus transmission process with large scale population. Compared with the earlier studies of modelling the epidemic dynamics, our model overcomes the following difficulties: i. The processes occurring at different scales, individual activities, rare events, sophisticated population structure must be altogether taken into account [45] in the real landscape. However, the traditional SEIR framework-based models or its variants based on differential equations [28, 47, 48] assuming a well-mixed population have difficulty to gain an insight of the model details. ii. The computational difficulty of modeling large scale population (7017144 people were taken into account in our study).

The hybrid model proposed in our study provides a reliable method that helps to analyze the impact of healthcare workers related policies and give efficient combination strategies

aiming to protect the high-risk populations against Ebola virus. Comparing our modeling result with other epidemic model which analyzed the intervention policies implemented for disease control and prevention, our work illustrates new findings in the presence of consistent conclusions with the earlier works. Ebola cases reported from Sierra Leone in HCWs represented a much higher estimated cumulative incidence in HCWs than in non-HCWs [49] which lead to a high efficiency of HCW-targeted vaccination [27]. Robert A et al. used a mathematical transmission model to explore the relative contribution of HCW and the community to transmission and claimed that ahead-of-time vaccination for 30% of HCW is effective for disease control [26] which is consistent to our conclusion given for the case that only vaccination strategies are considered. Besides, we also take into account the intervention of education and lead to the conclusion that vaccination of 10% HCW can be effective when pre-job training for HCW is also considered, corresponding to the statement that the achievement of one policy objective may preclude success with others [17]. Alexis Robert's et al. predicted that 10% of HCW getting prophylactic vaccination would result in 7685 cases and 3155 deaths and 30% of HCW getting vaccination would result in 5776 cases and 2358 [25] compared with our model output 6709 cases and 1987 deaths with 10% vaccination rate and 5425 cases and 1611 deaths with 30% vaccination rate. The difference between these two output mainly owing to the difference in the validation data. In our study, we do not differentiate between confirmed, probable and suspected cases and regard them as cases in our model for the SEIR model does not explicitly distinguish these three type of cases. Alexis Robert's model only considered the confirmed cases. However, our study come to a similar conclusion for vaccination strategies for the tend of the impact of vaccination are similar in these two models. Our model can be seen as an extension or a calibration for the earlier study.

We have identified several improvements that can be made which can be further studied. Firstly, the infected HCWs and the infected general population were assumed to be homogeneous in our study. In the real world, this assumption cannot be held fully which may lead to a bias between the model output and the real data. Secondly, the relation between vaccination policy and education policy is much more sophisticated in the real world than that in our model. The coupling relationship between these two policies can be further investigated.

To conclude, our hybrid simulation model successfully presents the dynamics of EVD transmission in Sierra Leone and evaluates the impact of different interventions. Our hybrid modeling and simulation method can be applied to understand the individual dynamics during epidemic spreading in large scale population. Besides, such models with heterogeneous human mobility can help inform disease control strategies for different infectious disease.

## Supporting information

**S1 File. Supporting data.**
(DOCX)

## Author Contributions

**Conceptualization:** Peiyu Chen, Wenhui Fan.

**Data curation:** Peiyu Chen, Xudong Guo.

**Formal analysis:** Peiyu Chen.

**Funding acquisition:** Peiyu Chen.

**Methodology:** Peiyu Chen, Wenhui Fan.

**Project administration:** Peiyu Chen, Wenhui Fan.

**Resources:** Peiyu Chen, Xudong Guo.

**Supervision:** Wenhui Fan, Xudong Guo.

**Validation:** Peiyu Chen, Xudong Guo.

**Visualization:** Peiyu Chen.

**Writing – original draft:** Peiyu Chen.

**Writing – review & editing:** Peiyu Chen, Wenhui Fan.

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
