## [Decision Letter · Decision Letter 0]

9 Mar 2021

PONE-D-20-38313

A hybrid simulation model to study the impact of combined interventions on Ebola epidemic

PLOS ONE

Dear Dr. Chen,

Thank you for submitting your manuscript to PLOS ONE. After careful consideration, we feel that it has merit but does not fully meet PLOS ONE’s publication criteria as it currently stands. Therefore, we invite you to submit a revised version of the manuscript that addresses the points raised during the review process.

Both reviewers agree that the presentation of the work needs more elaboration and the use of English should be considerably improved. One of the reviewers is particularly critical on the work, thus raising major issues about the model. The criticisms are mainly focused on the review of the literature and importantly  on the presentation of both models (mean field and agent-based one) and the lack of information concerning the statistical analysis of the results and the algorithms used for calibration purposes.

We look forward to receiving your revised manuscript.

Kind regards,

Constantinos Siettos, Ph.D.

Academic Editor

PLOS ONE

Journal Requirements:

2. Please ensure you have thoroughly discussed any potential limitations of this study within the Discussion section, including the potential impact of confounding factors.

"The authors would like to thank the financial support given by The National Key

Research and Development Program of China|Collective intelligence network

simulation and the experimental platform development (2017YFB1400105)."

4. Thank you for submitting the above manuscript to PLOS ONE. During our internal evaluation of the manuscript, we found significant text overlap between your submission and the following previously published works.

- https://doi.org/10.1371/journal.pone.0168127

- https://doi.org/10.1016/j.ecolmodel.2018.06.011

We would like to make you aware that copying extracts from previous publications, especially outside the methods section, word-for-word is unacceptable, even for works which you authored. In addition, the reproduction of text from published reports has implications for the copyright that may apply to the publications.

Please revise the manuscript to rephrase the duplicated text, cite your sources, and provide details as to how the current manuscript advances on previous work. Please note that further consideration is dependent on the submission of a manuscript that addresses these concerns about the overlap in text with published work.

Reviewers' comments:

Reviewer's Responses to Questions

**Comments to the Author**

1. Is the manuscript technically sound, and do the data support the conclusions?

Reviewer #1: Partly

Reviewer #2: Partly

2. Has the statistical analysis been performed appropriately and rigorously? 

Reviewer #1: N/A

Reviewer #2: I Don't Know

3. Have the authors made all data underlying the findings in their manuscript fully available?

Reviewer #1: No

Reviewer #2: Yes

4. Is the manuscript presented in an intelligible fashion and written in standard English?

Reviewer #1: Yes

Reviewer #2: No

5. Review Comments to the Author

Reviewer #1: The authors present a hybrid Agent-based model coupled with a compartmental SEIR model to describe the dynamics of the 2014-2015 Ebola in Sierra Leone. Based on the models, they study the effects of health care training and vaccination strategies.

Minor

- Line 16: should be Legrand not Legramd

- The authors use the term “system dynamics model” for SEIR models. The usual term used for such models is compartmental or mechanistic dynamical models. I don’t insist that they change the terminology, but this is more appropriate.

- The authors should elaborate more in the use of English. There are several syntactical errors.

Major

1. Line 18: “Furthermore, Tsanou et al. [9] studied the potential impact of 18environmental prophylactic vaccine” The word vaccine is misleading and should NOT be used in that way. Tsanou et al. studied the effect of environmental contamination and how adequate hygienic living conditions could affect the spread.

2. Line 20: following up the previous statement they state that “However, the stochastic nature of the movement of the individual is not considered explicitly with SDM for they only describe the mobility dynamics of sub-populations [10].” This sentence is vague and needs more elaboration. What do they mean with “for they only describe the mobility dynamics of sub-populations?” The model in Tsanou it is a compartmental model and not an individual-based model? This is what they mean?

3. Line 27. The authors state “However, it is tough to gain an insight of the epidemic dynamics 27with vole populations through ABM due to the massive calculations.”. The main difficulty when using ABM is not the massive calculations, but (1) the uncertainty in calibrating the many parameters and variables, the connection to the compartmental sub-group scale in a systematic way, the uncertainty in modeling the underlying high dimensional contact network. Computational cost is secondary.

4. A general comment about the review of the literature: There is a huge literature in epidemiology about agent-based models that have been used for various purposes. A relatively small subgroup of these have been developed for describing the Ebola epidemic in West Africa. They authors cite only a very small number of papers related to agent-based modelling but also not related to the epidemic of Ebola. Thus they have to elaborate more on the review of the existing literature, and they have to cite and discuss key papers that have introduced agent-based models that have focused on the Ebola dynamics. e.g. (in chronological order)

-

Siettos C, Anastassopoulou C, Russo L, Grigoras C, Mylonakis E. Modeling the 2014 Ebola Virus Epidemic - Agent-Based Simulations, Temporal Analysis and Future Predictions for Liberia and Sierra Leone. PLoS Curr. 2015;7:ecurrents.outbreaks.8d5984114855fc425e699e1a18cdc6c9. Published 2015

Merler S, Ajelli M, Fumanelli L, Gomes MF, Piontti AP, Rossi L, Chao DL, Longini IM Jr, Halloran ME, Vespignani A. Spatiotemporal spread of the 2014 outbreak of Ebola virus disease in Liberia and the effectiveness of non-pharmaceutical interventions: a computational modelling analysis. Lancet Infect Dis. 2015

Siettos CI, Anastassopoulou C, Russo L, et al. Forecasting and control policy assessment for the Ebola virus disease (EVD) epidemic in Sierra Leone using small-world networked model simulations. BMJ Open 2016;6:e008649. doi: 10.1136/bmjopen-2015-008649

Merler S, Ajelli M, Fumanelli L, Parlamento S, Pastore y Piontti A, Dean NE, et al. (2016) Containing Ebola at the Source with Ring Vaccination. PLoS Negl Trop Dis 10(11): e0005093

Srinivasan Venkatramanan, Bryan Lewis, Jiangzhuo Chen, Dave Higdon, Anil Vullikanti, Madhav Marathe. Using data-driven agent-based models for forecasting emerging infectious diseases, Epidemics,22, 2018,43-49.

5. The description of the modelling approach needs a lot of improvement. Actually the agent-based model is only described by a schematic and a table at the supporting information containing the transitions in the micro level.

6. Many of the parameters in the table of the SI appear as ad-hoc (70% possibility etc). These values can be justified.

7. The authors should describe this in detail. A prominent characteristic of agent-based models is the underlying social-transmission network. There is no such information in the presentation of the model. Without such a heterogeneity, i.e. when the interactions are uniform and random, an agent –based mode can be easily be substituted by a mean field model.

8. The same hold true for the calibration. There is no information about the algorithm that has been used to fit the parameters of the model and because of the fact that the agent-based model is not well described

Reviewer #2: Please refer to annotated attached file for my comments on the manuscript.

I believe the manuscript is worth investigating, but the English level is not good. The authors need to use short and precise sentences to express in a comprehensible manner their mind. Few technical questions mentioned on my report (the annotated attached manuscript) need to be answered rigorously for the overall manuscript to be more interesting and reader-friendly.

6. PLOS authors have the option to publish the peer review history of their article (what does this mean?). If published, this will include your full peer review and any attached files.

Reviewer #1: No

Reviewer #2: No

---

## [Author Response · Author response to Decision Letter 0]

21 Apr 2021

We have already uploaded the"Response to reviewers.pdf". All the required responds are covered in the file.

---

## [Decision Letter · Decision Letter 1]

25 May 2021

PONE-D-20-38313R1

A hybrid simulation model to study the impact of combined interventions on Ebola epidemic

PLOS ONE

Dear Dr. Chen,

Thank you for submitting your manuscript to PLOS ONE. After careful consideration, we feel that it has merit but does not fully meet PLOS ONE’s publication criteria as it currently stands. Therefore, we invite you to submit a revised version of the manuscript that addresses the points raised during the review process.

While both reviewers agree that you have accomodated most of their comments, and one of the two reviewers recommended the acceptance of your manuscript, the second reviewer has still some major comments that you have to adress before the final acceptance of you manuscript. In particular, the reviewer recommends that you should report the confidence intervals in your figures where appropriate; the second issue is about the way that you have calibrated your data.

We look forward to receiving your revised manuscript.

Kind regards,

Constantinos Siettos, Ph.D.

Academic Editor

PLOS ONE

Reviewers' comments:

Reviewer's Responses to Questions

**Comments to the Author**

1. If the authors have adequately addressed your comments raised in a previous round of review and you feel that this manuscript is now acceptable for publication, you may indicate that here to bypass the “Comments to the Author” section, enter your conflict of interest statement in the “Confidential to Editor” section, and submit your "Accept" recommendation.

Reviewer #1: (No Response)

Reviewer #2: All comments have been addressed

2. Is the manuscript technically sound, and do the data support the conclusions?

Reviewer #1: Partly

Reviewer #2: Yes

3. Has the statistical analysis been performed appropriately and rigorously? 

Reviewer #1: Yes

Reviewer #2: N/A

4. Have the authors made all data underlying the findings in their manuscript fully available?

Reviewer #1: Yes

Reviewer #2: Yes

5. Is the manuscript presented in an intelligible fashion and written in standard English?

Reviewer #1: Yes

Reviewer #2: Yes

6. Review Comments to the Author

Reviewer #1: The authors tried to address most of my comments in an adequate way. However, still the manuscript cannot be published in its current form. The authors should

1. work more on improving the quality of the figures (in the PDF version the quality of resolution is rather low) and importantly report and overlay the e.g. 95% confidence intervals in all relevant figures (figures for the cumulative ebola cases, deaths, etc, impact of pre-job training etc, impact of prophylactic vaccination.

2. Please make clear, what data have been used for calibration: on the cumulative or the new cases (at least for the deaths and recovered)?. It is known that a calibration based on long series of cumulative data can bias the confidence intervals of estimations. So it is better that the calibration is made on the new cases (see also the discussion in King et al. Avoidable errors in the modelling of outbreaks of emerging pathogens, with special reference to ebola, Proceedings of the Royal Society B: Biological Sciences 282 (1806) (2015) 20150347.

Reviewer #2: The authors made sufficient efforts to address satisfactorily all my concerns. The paper is now very nice, well writing and suitable for publication.

7. PLOS authors have the option to publish the peer review history of their article (what does this mean?). If published, this will include your full peer review and any attached files.

Reviewer #1: No

Reviewer #2: No

---

## [Author Response · Author response to Decision Letter 1]

7 Jun 2021

Please refer to the attachments for ''response for reviewers''.

---

## [Editor Report · Decision Letter 2]

21 Jun 2021

A hybrid simulation model to study the impact of combined interventions on Ebola epidemic

PONE-D-20-38313R2

Dear Dr. Chen,

We’re pleased to inform you that your manuscript has been judged scientifically suitable for publication and will be formally accepted for publication once it meets all outstanding technical requirements.

Kind regards,

Constantinos Siettos, Ph.D.

Academic Editor

PLOS ONE
---

## [Editor Report · Acceptance letter]

23 Jun 2021

PONE-D-20-38313R2 

A hybrid simulation model to study the impact of combined interventions on Ebola epidemic 

Dear Dr. Chen:

I'm pleased to inform you that your manuscript has been deemed suitable for publication in PLOS ONE. Congratulations! Your manuscript is now with our production department. 

Kind regards, 

on behalf of

Professor Constantinos Siettos 

Academic Editor

PLOS ONE